# Variability of Mercury Concentrations Across Species, Brand, and Tissue Type in Processed Commercial Seafood Products

**DOI:** 10.3390/toxics13060426

**Published:** 2025-05-23

**Authors:** Kylie D. Rock, Shriya Bhoothapuri, Emanuel Lassiter, Leah Segedie, Scott M. Belcher

**Affiliations:** 1Center for Environmental and Health Effects of PFAS, Department of Biological Sciences, North Carolina State University, Raleigh, NC 27695, USA; smbelch2@ncsu.edu; 2Department of Biological Sciences, Clemson University, 134 Long Hall, 230 Parkway Drive, Clemson, SC 29634, USA; 3Department of Biological Sciences, North Carolina State University, 127 David Clark Labs Campus Box 7617, Raleigh, NC 27607, USA; sbhooths22@gmail.com (S.B.);; 4Mamavation, 23 Corporate Plaza Drive, Suite 150-88, Newport Beach, CA 92660, USA; leah@mamavation.com

**Keywords:** mercury, seafood, canned food, fish, tuna

## Abstract

Mercury (Hg) is a global health concern due to its prevalence, persistence, and toxicity. Numerous studies have assessed Hg concentrations in seafood, but variability in reported concentrations highlights the need for continued monitoring and stricter regulations. We measured total Hg (tHg) in 148 pre-processed, packaged seafood products purchased in Raleigh, North Carolina, using thermal decomposition–gold amalgamation atomic absorption spectrophotometry. Products were grouped into three categories based on trophic ecology and physiology: (1) tuna, (2) other bony fish, and (3) shellfish and squid. Among tuna, albacore had the highest average tHg (396.4 ng/g ± 172.1), while yellowfin had the lowest (68.3 ng/g ± 64.7). Herring (54.0 ng/g ± 23.2) and crab (78.2 ng/g ± 24.1) had the highest concentrations in the other two groups. One can of albacore exceeded the FDA action level of 1 part per million (1.3 ppm or 1300 ng/g). Brand differences were significant for both albacore and light tuna, with Brand 1 consistently showing higher Hg levels. Comparisons to FDA data (1990–2012) suggest Hg concentrations in tuna have remained stable over the past two decades. This study underscores the variability of Hg concentrations across species and brands and the need for continued monitoring to protect consumers.

## 1. Introduction

According to the World Health Organization (WHO), mercury (Hg) is among the top 10 chemicals of global public health concern due to its prevalence, persistence, ability to bioaccumulate, and toxicity [1,2,3,4,5,6]. Worldwide, fish and shellfish serve as an important dietary source of lean protein and other nutrients, including omega-3 polyunsaturated fatty acids, vitamin B12, and vitamin D, which confer important long-term benefits for health and development [7,8,9,10,11]. However, seafood also contains Hg, primarily in the form of methylmercury (MeHg), and even at low concentrations, MeHg can have adverse health effects with toxicity observed in the kidneys, nervous, cardiovascular, digestive, and immune systems [12,13,14,15,16,17,18]. Because most human exposure to MeHg occurs through the consumption of seafood and global seafood demand continues to rise, there is a critical need for continued monitoring of Hg concentrations in commercial fish and shellfish harvested from marine waters [12,19,20,21]. Such efforts will better inform exposure and risk estimates to help consumers make informed decisions about the types and quantities of seafood they consume [22].

In the United States, the Food and Drug Administration (FDA) and Environmental Protection Agency (EPA) have issued a consumption advisory for particularly vulnerable individuals, specifically during fetal, infant, and childhood development, warning women who might become pregnant, are pregnant, or are breastfeeding and children to avoid eating specific species of fish shown to have high concentrations of Hg [11,23]. Given the health benefits associated with seafood consumption, particularly for growth and development, the FDA and EPA issued “Advice about Eating Fish”, featuring a chart that can be used as a reference to make informed choices about fish and shellfish products that are nutritious and minimize Hg exposure [11]. Documents such as this are useful tools for raising public awareness and communicating safe alternatives, but will require continued monitoring of frequently consumed and commercially available products to be successful.

Tuna is among the top five species of seafood consumed in the U.S., with annual consumption estimated at 400,000 metric tons [24,25]. Tuna are large, long-lived pelagic predators that occupy a high trophic position in marine ecosystems and tend to concentrate Hg in their tissues, making them a focal species for evaluating human exposure. Tuna is sold in fresh, frozen, and canned varieties, but due to its availability and affordability, canned tuna accounts for nearly 70% of tuna consumption in the U.S. [24,26,27]. Numerous studies have evaluated Hg concentrations in commercial tuna products, demonstrating a high degree of intra- and interspecies variability and potentially brand-specific differences in Hg concentrations [27]. However, fewer studies have evaluated these trends in alternative seafood products, including other bony fish, shellfish, and squid. Furthermore, studies have reported Hg concentrations in canned tuna that are above the FDA action level of 1 part per million (PPM), a regulatory standard that allows the FDA to take legal action to remove products from the market that exceed this limit, raising concerns about the reliability of exposure estimates and consumption advice [28,29].

In the present study, we purchased a variety of packaged fish products, including tuna, other bony fish, shellfish, and squid, from a wide range of brands to measure total Hg (tHg) concentrations in readily available commercial products. We tested multiple hypotheses in this study. First, because of differences in trophic ecology (e.g., predators or primary consumers) and physiology (e.g., vertebrates or invertebrates), and data from prior Hg monitoring studies, we hypothesized that canned tuna would have the highest concentrations of Hg compared to other bony fish, shellfish, and squid [30]. Further, we hypothesized that physiological differences in vertebrates and invertebrates contribute to differences in the bioaccumulation of Hg. Second, previous studies have demonstrated that Hg concentrations in canned tuna can vary by brand, likely due to differences in where the fish are caught and/or processed. Therefore, we hypothesized that Hg concentrations in canned tuna would differ by brand, packaging material (can, pouch, glass), and/or packaging medium (water or oil). Finally, even though steps are being taken to reduce Hg emissions, we hypothesized that Hg concentrations in canned tuna would closely approximate concentrations previously reported by the FDA.

## 2. Methods

### 2.1. Study Design

Packaged fish products were purchased in April 2022 from various grocery stores to sample from a wide range of brands, in Raleigh, North Carolina, United States. Products were purchased in groups of 3 or more (*n* = 3–44) and analyzed for differences in total Hg (tHg) concentration based on species, brand, packaging (can, pouch, glass), and packaging medium (water or oil; Appendix A). A variety of species were analyzed in this study, including albacore tuna (*Thunnus alalunga*), skipjack tuna (*Katsuwonus pelamis*), yellowfin tuna (*Thunnus albacares*), anchovies (*Engraulis encrasicolus*), herring (*Clupea harengus*), mackerel (*Scomber scombrus*), sardines (*Sardina pilchardus*), salmon (*Oncorhynchus gorbuscha*), trout (*Oncorhynchus mykiss*), clams (*Amaura delesserti*), crab (*Portunus pelagicus*), oysters (*Crassostrea gigas*), scallop (*Placopecten magellanicus*), and squid (exact species unknown).

### 2.2. Sample Preparation

Packages were opened inside a hood, liquid was drained, and a subsample of seafood was placed in 7 mL polypropylene tubes pre-filled with 2.8 mm ceramic beads (Thermo Fisher Scientific, Waltham, MA, USA; 15-340-157). Samples were homogenized for 3 min using a Fisherbrand^TM^ bead mill homogenizer equipped with a 7 mL tube carriage (Thermo Fisher Scientific; 15-340-164). After homogenization, samples were briefly centrifuged at 2000 relative centrifugal field (rcf) for 1 min. Each homogenized sample was aliquoted into 2 mL polypropylene tubes. One aliquot was stored at −80 °C until analysis, and one was freeze-dried using a FreeZone 4.5 L benchtop freeze dryer (Labconco, Kansas City, MO, USA). Tissue was freeze-dried in order to compare samples statistically across brand, packaging material, and packaging medium in order to remove moisture content as a confounding variable. Freeze-dried samples are referenced throughout the paper as dry weight (DW).

### 2.3. Total Mercury Analysis

Mercury analysis was performed between 21 April 2022 and 14 June 2022 using a MA-3000 mercury analyzer (Nippon Instruments, Osaka, Japan) following guidance outlined in US EPA methods 7473 and as previously described [30,31]. Each experimental sample was thawed, and approximately 10 mg subsamples were weighed at room temperature in a ceramic weigh boat. Total Hg (tHg) concentrations were determined in duplicate using a linear equation derived from a calibration curve that ranged from 0.1 to 1000 ng Hg (r^2^ = 0.99). The aqueous standards for the calibration curve were prepared by serial dilution of a certified HgCl_2_ standard in 2% HCl (1000 ± 5 µg/mL; lot# 1923928, AGS Scientific, Bryan, TX, USA). National Institute of Standards and Technology Standard Reference Material (SRM) 1947, Lake Michigan fish tissue (254 ng/g ± 0.0005), was used to validate the tHg measurements. Replicate measurements of SRM 1947 had a coefficient of variation of 5.6% (*M* = 252.4 ng/g, *SD* = 14.1, *n* = 27). Method blanks were run on each day of analysis, between every 10 samples, as quality control, and the limit of detection, defined as LOD = mean_blanks_ + 3 (SD_blanks_), was 0.002 ng (*n* = 27).

### 2.4. Comparison to Historical Data—FDA 1990–2012

Because tuna is one of the most widely consumed seafood products in the US, with albacore and light tuna being the most prevalent options in grocery stores, we compiled the previously published data on canned albacore and light tuna from the FDA national marine fisheries survey of mercury in commercial fish spanning from 1990 to 2010 [23]. The publicly available FDA data were sorted and calculated as follows: Data were sorted by year, and the mean and geometric mean mercury concentrations for albacore and light tuna were calculated [23]. From our data set, we performed the same calculations using products specifically labeled albacore or light tuna. It is important to note that the analytical method used to measure Hg concentrations in the FDA study, inductively coupled plasma mass spectrometry (ICP-MS), was different from the one used in this study. Where non-detects were reported, in other words, Hg concentrations below the detection level of 0.01 PPM, we assigned an interpolated concentration of LOQ/√2 (Appendix A).

### 2.5. Statistical Analysis

All data were analyzed using Prism (version 10.0, GraphPad, La Jolla, CA, USA). The coefficient of variation for all analyzed samples, except for one crab sample, was ≤15% (Appendix A). A Shapiro–Wilk test was used to assess the normality of all data. Because of differences in trophic ecology (e.g., predators or primary consumers) and physiology (e.g., vertebrates or invertebrates), we hypothesized that Hg concentrations would be significantly different across the three broad groups of samples we purchased, tuna, other bony fish, and shellfish and squid, with Hg concentrations being much higher in albacore tuna than in the other two groups. Therefore, to investigate concentration differences among seafood types, we ran three separate one-way analyses of variance (ANOVAs) with Tukey’s post-hoc tests for multiple comparisons for (1) tuna, (2) other bony fish, and (3) shellfish and squid. All data were log_10_-transformed for statistical analysis. Log-transformed DW tHg values for light and albacore tuna were also analyzed using a univariate general linear model to investigate the impact of brand, packaging, and packaging medium on tHg concentration. If a significant effect was identified, a follow-up one-way ANOVA was performed.

When comparing our data set to previously published FDA data, we utilized our wet weight (WW) tHg values, as this is what was reported by the FDA. Because none of the groups passed normality, and we had some tHg values less than 1, WW tHg concentrations were transformed using log_10_ [32,33,34]. A standard one-way ANOVA was used to assess differences in tHg concentrations over the years, for the FDA study (2000–2010) and our study (2022). For all statistical analyses, the minimal level of statistical significance for differences in values among or between groups was considered *p* ≤ 0.05.

## 3. Results

### 3.1. Total Hg Concentrations in Commercial Fish and Shellfish

An analysis of tHg in packaged fish products showed mean concentrations ranging from 7.8 ng/g (*SD* = 0.2) in clams to 396.4 ng/g (*SD* = 172.1) in albacore tuna (Appendix A). A significant difference in tHg concentrations was observed across the different types of tuna (*F*(3, 91) = 74.23; *p* ≤ 0.001; *η*^2^ = 0.71), with albacore having significantly higher tHg concentrations than all other tuna (*p* < 0.001) and light and skipjack tuna having significantly higher tHg concentrations than yellowfin tuna (*p* < 0.001; *p* < 0.001; Figure 1A). A significant difference in tHg concentrations was also found among the other bony fish (*F*(5, 26) = 11.91; *p* ≤ 0.001; *η^2^* = 0.70). Concentrations of tHg were significantly higher in herring than salmon (*p* < 0.001), sardines (*p* < 0.001), anchovies (*p* = 0.001), and trout (*p* < 0.001), and tHg in mackerel was significantly higher than that in sardines (*p* = 0.02) and trout (*p* = 0.009; Figure 1B). Finally, a significant difference in tHg concentrations was observed for shellfish and squid (*F*(4, 16) = 26.86; *p ≤ 0*.001; *η*^2^ = 0.87), where tHg concentrations in crabs were significantly higher than those in clams (*p* < 0.001), oysters (*p* < 0.001), scallops (*p* < 0.001), and squid (*p* = 0.02), and tHg concentrations in squid were significantly higher than those in clams (*p* = 0.002) and oysters (*p* = 0.04; Figure 1C).

### 3.2. Comparison of tHg Concentrations to Federal Standards

In addition to within-study comparisons, we determined the percentage of samples with tHg concentrations above federal standards based on wet weight for each type of food analyzed (Appendix A). Notably, samples of albacore tuna exceeded the FDA action level of 1 PPM (i.e., 1000 ng/g; 2%), the EPA screening value for recreation of 0.4 PPM (i.e., 400 ng/g; 9%), and the EPA screening value for subsistence of 0.049 PPM (i.e., 49 ng/g; 100%; Appendix A). Additionally, samples of light tuna (89%), skipjack tuna (100%), yellowfin tuna (46%), herring (50%), and crab (83%) exceeded the EPA screening value for subsistence (Appendix A).

### 3.3. Total Mercury Concentrations Varied by Brand and Meat Type

tHg concentrations were also analyzed via DW in order to compare concentrations across brands, package types, packaging media, and meat types (e.g., white vs. lump) without moisture content confounding the results. In tuna, we identified a significant overall effect of brand on log-transformed tHg DW concentrations in albacore (*F* = 4.04; *p* = 0.002; *η*^2^ = 0.51, Figure 2A, Appendix A) and light tuna (*F* = 4.07; *p* = 0.01; *η*^2^ = 0.52; Figure 2B, Appendix A). Concentrations of tHg in albacore tuna from brand 1 were significantly higher than those from brands 2 (*p* = 0.01), 4 (*p* < 0.001), 5 (*p* = 0.01), and 6 (*p* = 0.002; Figure 2A). Among light tuna, tHg concentrations from brands 1 (*p* < 0.001) and 2 (*p* = 0.001) were significantly higher than those from brand 8 (Figure 2B). No significant overall effect of packaging or packaging medium was observed. In addition to tuna, a significant effect of brand was observed for herring, with significantly higher concentrations observed in brand 15 compared to brand 18 (*p* = 0.001; Figure 2C,D). Finally, when comparing tHg concentrations in shellfish and squid, we found that lump crab meat had significantly higher concentrations than white crab meat (*p* = 0.02; Figure 2E,F).

### 3.4. Comparison of tHg Concentrations to the FDA 1990–2012 Hg in Commercial Fish Survey

Overall, our mean and geometric mean values are comparable to those previously reported by the FDA (Appendix A); however, a significant difference in log_10_(X + 1)-transformed tHg concentrations across years was identified for albacore tuna (*F*(7, 478) = 3.68; *p* ≤ 0.001; *η*^2^ = 0.05) and light tuna (*F*(8, 518) = 14.77; *p* ≤ 0.001; *η*^2^ = 0.19). For albacore tuna, tHg concentrations were higher in 2022 compared to 2004 (*p* = 0.01), 2005 (*p* = 0.02), and 2007 (*p* = 0.02; Figure 3A). For light tuna, tHg concentrations were significantly higher in 2022 compared to 2005 (*p* < 0.001; Figure 3B).

## 4. Discussion

Canned tuna remains a major component of the American diet due to its convenience, affordability, taste, and health benefits. However, compared to other readily available canned seafood products, tuna, particularly albacore tuna, has some of the highest Hg concentrations [23,27,28]. Studies have established links between frequent fish consumption, increased Hg exposure, and adverse health outcomes [35,36,37,38]. Although the Minamata Convention on Mercury is an unprecedented step towards protecting human and environmental health from Hg emissions, it will be years before we see significant benefits from these efforts because of the prevalence of Hg in the environment. Consequently, continued monitoring of frequently consumed fish products and dissemination of this information to the public are needed to reduce the risks associated with fish consumption.

In this study, we analyzed tHg concentrations in 148 ready-to-eat packaged fish products from 23 different brands that were available in local grocery stores in Raleigh, North Carolina. All the samples analyzed had measurable levels of tHg, with the highest concentrations measured in albacore tuna. Similar to previous reports, mean concentrations of tHg in albacore tuna were higher than those in light tuna, reported to predominantly be composed of skipjack and yellowfin tuna [23,27,28]. Yellowfin tuna had the lowest Hg concentrations of all the tuna tested. These trends can partially be attributed to differences in size, foraging, and growth rates. For example, albacore tuna are more piscivorous than skipjack and yellowfin tuna, which mainly feed on invertebrates, making them more likely to accumulate Hg in their tissues [27,39,40]. Of the other bony fishes that were analyzed in this study, anchovies, sardines, and trout had the lowest tHg concentrations. Collectively, these findings largely agree with previous surveys of Hg in fish (Appendix A) and highlight some alternative options to tuna, particularly albacore tuna, that reduce the risk of Hg exposure.

Interestingly, among the invertebrates, crab stood out as having the highest tHg concentrations, and canned lump crab meat had higher tHg than white meat. Lump crab meat is typically composed of a mixture of white meat collected from the body of the crab and claw and leg meat. Protein distribution within organisms varies by muscle location and function and can impact differences in Hg accumulation across the body due to its affinity for thiol (i.e., R-SH) groups [27,41,42,43,44,45]. Previous studies of Hg accumulation and distribution across muscle types in tuna have demonstrated that dark muscle, which has higher metabolic activity and a higher rate of muscle fiber development, has higher Hg concentrations than white muscle [43,46]. Likewise, differences in protein composition and metabolic activity in the darker leg and claw meat compared to white body meat may explain the observed differences in tHg concentrations between canned lump and white crab. Additional studies are needed, especially in species where the consumption of “dark” meat is preferred, such as crab and lobster, to better communicate these risks to regulatory agencies and the public.

In addition to fish consumption advisories, the FDA has established a legal action level of 1.0 PPM, and the EPA has set screening values for fish consumed by recreational fishers (0.4 PPM) and subsistence fishers (0.049 PPM; Appendix A). Notably, one (3%) of our cans of albacore tuna exceeded the FDA action limit, reaching 1.3 PPM, and four (9%) samples fell above the EPA advisory for recreational fish consumption. Given our relatively small sample size, relative to the amount of canned tuna that is consumed in the U.S., more work is needed to assess the extent to which people may be exposed to Hg concentrations above the FDA action level. Similar findings from canned tuna purchased in Las Vegas, Nevada, U.S., were reported back in 2009 [28]; however, none of the FDA studies report a max concentration in canned albacore tuna above their established action level of 1 PPM [21]. Differences in analysis methods may contribute to discordance in results across studies, emphasizing the need to standardize techniques when establishing global monitoring programs [22,45]. Continued detection of Hg concentrations that exceed these federal limits raises concerns about the regulation of the canned tuna industry and the safety of sensitive populations, including pregnant women, infants, and children.

Concentrations of tHg in albacore and light tuna were found to be different across brands. In particular, brand 1 had some of the highest concentrations of tHg for both albacore and light tuna. These differences could be due to a variety of factors, including (1) catch location, (2) the age and size of the fish caught, and (3) how the fish are processed and packaged [47,48]. Hg concentrations in aquatic biota are greatly dependent upon the bioavailability of Hg in their surrounding environment. This has recently been demonstrated in wild yellowfin and bluefin tuna, demonstrating that geographic origin is a critical factor in determining Hg concentration in fish [29,49]. Although the age and size of fish are not reported to the public, several companies provide information about where the fish are caught. For this reason, we had planned to incorporate catch location in our analysis. However, inconsistencies in reporting, including the use of vague descriptors (e.g., North and South Pacific) or a complete lack of location information, made this not possible. Identifying a significant effect of brand, but no effect of packaging material or medium, may suggest that catch location is a stronger determinant of Hg concentrations than how the fish are processed and packaged. Increased transparency and standardization in the reporting of fish size, age, and catch location would be helpful for monitoring changes in global Hg pollution and the human health risk associated with tuna consumption.

Compared to previous monitoring studies conducted by the U.S. FDA between 2000 and 2009, Hg concentrations in ready-to-eat albacore and light tuna analyzed in this study were similar. More recent studies by the FDA have reported concentrations of Hg in a variety of foods, including canned tuna; however, the current summary reports do not provide specific details on the type of tuna or the raw values for performing statistical comparisons [50]. For both albacore and light tuna, FDA tHg values highlighted decreasing trends in concentration between 2000 and 2005 for light tuna and 2007 for albacore tuna. These declines may reflect efforts that have been made to reduce Hg emissions in North America and Europe; however, other factors, including changes in stock fish size distribution, may play a role [51]. Unfortunately, emissions in Central America, South Asia, and Eastern Africa have increased, and global emissions have remained relatively stable since the 1980s, which has likely contributed to consistency in Hg concentrations in tuna over the last couple of decades.

Hg has been a global contaminant of concern for decades, posing significant environmental and public health risks due to its prevalence, persistence, and toxicity. The findings from the present study are consistent with data found in the literature, demonstrating that the accumulation of Hg in frequently consumed commercial fish products has not changed much over the last two decades despite increased awareness and concern regarding Hg toxicity. Furthermore, our findings highlight that Hg concentrations in canned albacore tuna can exceed the FDA action level, which is particularly concerning for at-risk populations such as women, infants, and children. Stricter controls and routine monitoring efforts are needed at production sites before commercial tuna products are placed on the market. The Minamata Convention on Mercury is a novel effort towards mitigating human and environmental health risks associated with Hg exposure. Until the convention establishes a long-term global monitoring program, efforts are needed to evaluate Hg concentrations and ensure the safety of the tuna that we consume.

## Figures and Tables

**Figure 1 toxics-13-00426-f001:**
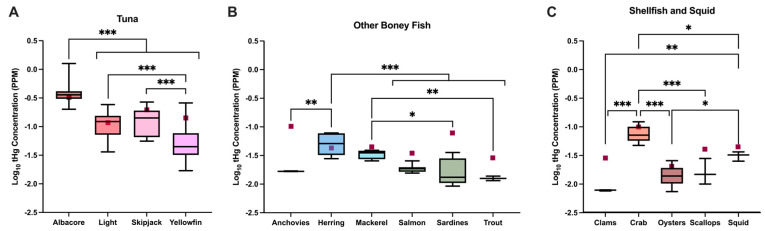
tHg concentrations vary significantly across species. Total Hg concentrations, wet weight (WW), in (**A**) albacore tuna (n = 44), light tuna (n = 27), skipjack tuna (n = 11), and yellowfin tuna (n = 13), (**B**) anchovies (n = 3), herring (n = 6), mackerel (n = 4), salmon (n = 9), sardines (n = 7), and trout (n = 3), and (**C**) clams (n = 3), crab (n = 6), oysters (n = 6), scallops (n = 3), and squid (n = 3); one-way ANOVA, * *p* < 0.05, ** *p* < 0.01, *** *p* < 0.001. Maroon squares indicate average concentration as reported by Karimi et al., 2012 [22]. Graphs indicate mean, min, max.

**Figure 2 toxics-13-00426-f002:**
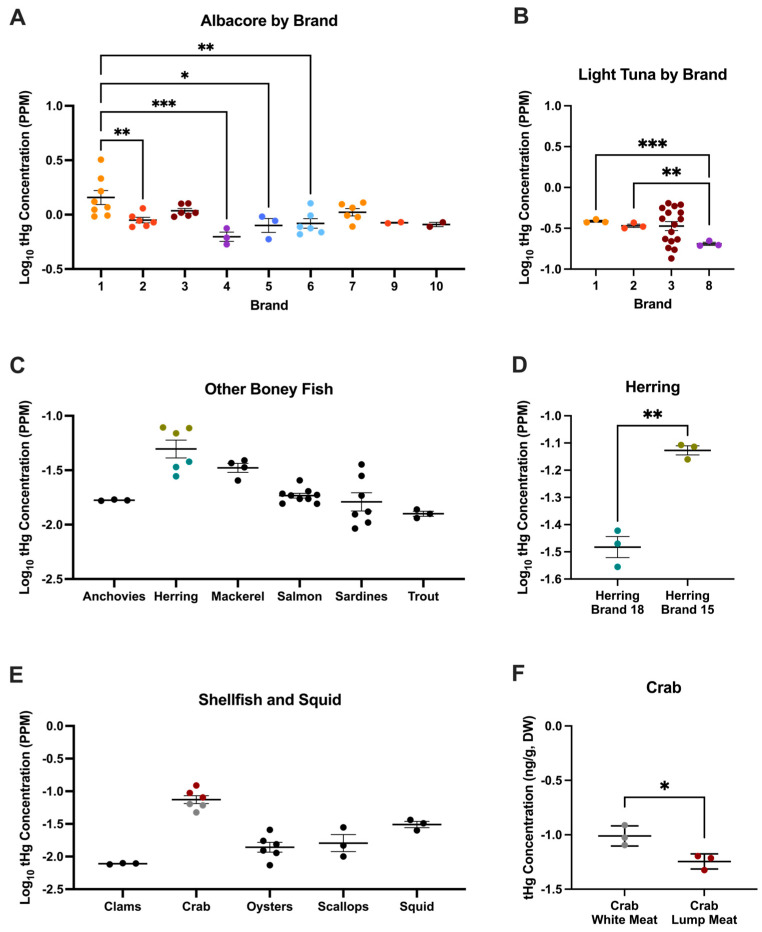
tHG concentrations in this study differ across brands and meat types. One-way ANOVA of log_10_-transformed tHg concentrations in (**A**) albacore tuna, (**B**) light tuna, (**C**) other bony fish, (**D**) *t*-test between herring from Brand 18 and Brand 15, One-way ANOVA in (**E**) shellfish and squid, and (**F**) *t*-test between crab white meat and lump meat; one-way ANOVA and *t*-test, * *p* < 0.05, ** *p* < 0.01, *** *p* < 0.001. Graphs indicate mean ± SD.

**Figure 3 toxics-13-00426-f003:**
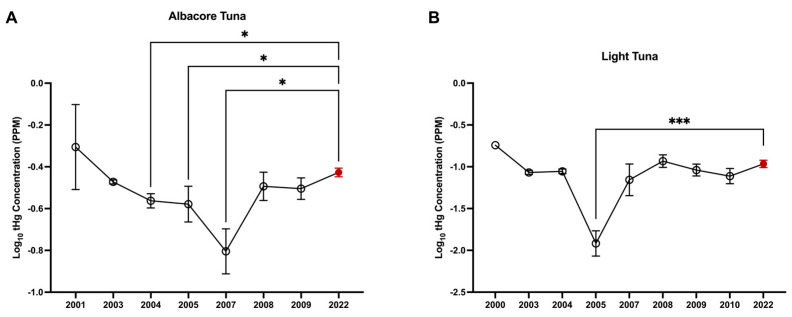
tHG concentrations in this study are comparable to those previously reported by the FDA. One-way ANOVA of log_(x+1)_-transformed tHg concentrations in (**A**) albacore tuna and (**B**) light tuna; one-way ANOVA, * *p* < 0.05, *** *p* < 0.001. Data shown for the years 2000–2010 were compiled from the FDA’s previously published data set (black data points), while the data shown for 2022 represent the data analyzed in this study (red data point). Graphs indicate mean ± SD.

## Data Availability

The original contributions presented in this study are included in the article/Appendix A. Further inquiries can be directed to the corresponding author.

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
