# Peer review of "Variability of Mercury Concentrations Across Species, Brand, and Tissue Type in Processed Commercial Seafood Products"

_toxics, 2025, doi:10.3390/toxics13060426_

Round 1
Reviewer 1 Report
Comments and Suggestions for Authors
Summary
The authors report a cross-sectional survey of mercury in various seafood products purchased from stores in one US region
General Comments
Overall, the authors report a well-justified study, with adequate and reproducible methods, generally reasonable statistical analysis, and valid interpretation of their data. What follows are a few comments to improve the analysis and writing of the work.
Consider plotting data in log scale. In most cases (Fig 1 and 2) the authors plot their data in what I think linear scale even though they are (correctly) using log10 transformed data for statistical analysis. I would suggest they then plot the data in the log transformed scale. This is particularly obviously necessary in Fig 2, D, where they use an inset panel to display low values. Log transformation would make those visible. Most graphing programs can simply scale the axes in log if they want to retain the natural numbers.
Consider annotation plots with multiple comparisons not reference comparisons. In the plots I would suggest reporting the results of the multiple comparisons tests withing each group, rather than comparisons to reference classes like albacore. In particular, this would allow one to justify from a figure the discussion point in line 261 and yellowfin tuna had the lowest concentrations of the tuna. Currently that’s buried in the text, and all we see on the figure is that it is lower than albacore.
Line Item Comments
L24, 28, elsewhere: It is good the paper typically reports a standard unit of ng/g. So writing part per million or PPM is OK, but then I’d suggest somewhere reminding readers what this means in ng/g. This is most critical for the abstract to be easily understood, and L136 where the LOD is specified in PPM.
Intro: Another metric one might want to compare the seafood types by is relative consumption in the US.
L108: Why freeze dried? That doesn’t seem to be used.
L158. I don’t understand why the X+1 was needed for the log10 transformation. I understand you cannot take the log10 of values of 0. But in the text you specify some tHg values less than 1. Log10 does just find with decimal values, like 0.1 would become -1 in log10 scale.
168-176. Given all of this text, this is where reporting the multiple comparisons values on the graph might make things easier to understand
Figure 1. There are some technical issues with this figure (I think). I don’t understand the Y axis scale. It appears almost linear, but then why is the lowest tick 0.049 not 0.05 which would be exactly linear. Though based on above, I’d at least consider making it log scale. It seems like the EPA screening value recreational may be in the wrong spot, as it is plotted at 0.40, and line 197 states the value is 0.05 ppm (and another case were ppm to ng/g could be clarified). One should also define what the box plot box widths are, typically the interquartile range. And I’d suggest putting the real pvalues not letters, but I guess that’s discretionary.
L217. Seems to repeat the figure legend.
Author Response
Summary
The authors report a cross-sectional survey of mercury in various seafood products purchased from stores in one US region
General Comments
Overall, the authors report a well-justified study, with adequate and reproducible methods, generally reasonable statistical analysis, and valid interpretation of their data. What follows are a few comments to improve the analysis and writing of the work.
Consider plotting data in log scale. In most cases (Fig 1 and 2) the authors plot their data in what I think linear scale even though they are (correctly) using log10 transformed data for statistical analysis. I would suggest they then plot the data in the log transformed scale. This is particularly obviously necessary in Fig 2, D, where they use an inset panel to display low values. Log transformation would make those visible. Most graphing programs can simply scale the axes in log if they want to retain the natural numbers.
We appreciate your feedback on this and agree. We have converted all figures to log scale.
Consider annotation plots with multiple comparisons not reference comparisons. In the plots I would suggest reporting the results of the multiple comparisons tests withing each group, rather than comparisons to reference classes like albacore. In particular, this would allow one to justify from a figure the discussion point in line 261 and yellowfin tuna had the lowest concentrations of the tuna. Currently that’s buried in the text, and all we see on the figure is that it is lower than albacore.
We have adjusted our plots to reflect the results of all multiple comparisons and separated graphs by “fish type” to increase clarity.
Line Item Comments
L24, 28, elsewhere: It is good the paper typically reports a standard unit of ng/g. So writing part per million or PPM is OK, but then I’d suggest somewhere reminding readers what this means in ng/g. This is most critical for the abstract to be easily understood, and L136 where the LOD is specified in PPM.
This has been emphasized in the abstract as well as the results (see “Comparison of tHg Concentrations to Federal Standards”).
Intro: Another metric one might want to compare the seafood types by is relative consumption in the US.
L108: Why freeze dried? That doesn’t seem to be used.
The samples were freeze dried to remove the confounding variable of water content to do comparisons across brands and meat types. This has been clarified on line 109: “Tissue was freeze dried in order to compare samples statistically across brand, packaging material, and packaging medium in order to remove moisture content as a confounding variable. Freeze dried samples are referenced throughout the paper as dry weight (DW).”
L158. I don’t understand why the X+1 was needed for the log10 transformation. I understand you cannot take the log10 of values of 0. But in the text you specify some tHg values less than 1. Log10 does just find with decimal values, like 0.1 would become -1 in log10 scale.
We have adjusted this method and only used log10 for all data analysis.
168-176. Given all of this text, this is where reporting the multiple comparisons values on the graph might make things easier to understand.
Agreed. And as mentioned above we have corrected this.
Figure 1. There are some technical issues with this figure (I think). I don’t understand the Y axis scale. It appears almost linear, but then why is the lowest tick 0.049 not 0.05 which would be exactly linear. Though based on above, I’d at least consider making it log scale. It seems like the EPA screening value recreational may be in the wrong spot, as it is plotted at 0.40, and line 197 states the value is 0.05 ppm (and another case were ppm to ng/g could be clarified). One should also define what the box plot box widths are, typically the interquartile range. And I’d suggest putting the real p values not letters, but I guess that’s discretionary.
We have separated figure 1 into three different graphs all of which have been changed to log scale. Thank you for catching the typo in the text. For EPA, recreational screening value is 0.4PPM and subsistence screening value is 0.049PPM. This has been corrected.
L217. Seems to repeat the figure legend.
This has been removed.
Reviewer 2 Report
Comments and Suggestions for Authors
The manuscript “Variability of Mercury Concentrations Across Species, Brand, 2and Tissue Type in Processed Commercial Seafood Products” is a potentially interesting paper for the study of this toxic element in fish products to assess food safety risks.
Unfortunately, the work does not present any innovative information but is a simple monitoring of the concentration of mercury in some foods. In lines 83-85 the Authors hypothesized that canned tuna would have the highest concentrations of Hg compared to other boney fish, shellfish, and squid and that physiological differences in vertebrates and invertebrates contribute to differences in bioaccumulation of Hg. However, there is already a large literature on this topic, molluscs, cephalopods and crustaceans contribute little to dietary exposure to mercury, while predatory and large fish such as tuna and swordfish are the species that accumulate most mercury.
Second, the Authors affirmed that previous studies have demonstrated that Hg concentrations in canned tuna can vary by brand, likely due to differences in where the fish are caught and/or processed. Therefore, they hypothesized that Hg concentrations in canned tuna would differ by brand, packaging material (can, pouch, glass). To verify this hypothesis, the Authors should have ensured that the origin of the products was the same in order to eliminate this variable, but no information has been provided on this aspect.
In the paper the mercury values are expressed ng/g while the FDA, to which the Authors refer, expresses the mercury concentrations in PPM (micrograms/g or mg/kg) as well as the European regulation on maximum levels of metals in some food products. This makes the reading of the results unclear.
There is also great confusion in the sample size, for example for albacore tuna table S4 reports 44 samples but the sampling was carried out by taking 3 units (assumed to be of the same product) where possible (see line 97). This count leads to altered values that do not correspond to a variety of different samples.In my opinion the sample size for some species (oysters, squid, clams) is really too low to be able to use average values
Regarding other results on similar products, among the references cited in the manuscript, only the one by Karimi et al., 2012 is considered in the tableS9.
Figure 2 brings together the comparison between brands (A and B) and different types of products and this generates confusion.
Figure 3 represents the trend of mercury concentrations detected by the FDA to which the result of the study is improperly added.
Author Response
The manuscript “Variability of Mercury Concentrations Across Species, Brand, and Tissue Type in Processed Commercial Seafood Products” is a potentially interesting paper for the study of this toxic element in fish products to assess food safety risks.
Unfortunately, the work does not present any innovative information but is a simple monitoring of the concentration of mercury in some foods. In lines 83-85 the Authors hypothesized that canned tuna would have the highest concentrations of Hg compared to other boney fish, shellfish, and squid and that physiological differences in vertebrates and invertebrates contribute to differences in bioaccumulation of Hg. However, there is already a large literature on this topic, mollusks, cephalopods and crustaceans contribute little to dietary exposure to mercury, while predatory and large fish such as tuna and swordfish are the species that accumulate most mercury.
Yes, it is true that this relationship has been shown many times, and from many different regions of the world. Therefore, we posed multiple hypotheses for this study, including ones supported by previously published data.
Second, the Authors affirmed that previous studies have demonstrated that Hg concentrations in canned tuna can vary by brand, likely due to differences in where the fish are caught and/or processed. Therefore, they hypothesized that Hg concentrations in canned tuna would differ by brand, packaging material (can, pouch, glass). To verify this hypothesis, the Authors should have ensured that the origin of the products was the same in order to eliminate this variable, but no information has been provided on this aspect.
We attempted to incorporate catch location in our assessment, however as we cover in the discussion most companies were inconsistent in how they reported catch location, using vague terminology like north or south pacific or not reporting catch location at all. Therefore, we were not able to confidently incorporate the impact of catch location into our analyses and instead broadly tested for impacts of brand on Hg concentrations.
In the paper the mercury values are expressed ng/g while the FDA, to which the Authors refer, expresses the mercury concentrations in PPM (micrograms/g or mg/kg) as well as the European regulation on maximum levels of metals in some food products. This makes the reading of the results unclear.
To make this more clear to the readers we have incorporated what the relevant concentration in ng/g for a given ppm is in both the abstract and results. Ex: “One can of albacore exceeded the FDA action level of 1 part per million (1.3 ppm or 1300 ng/g). “
There is also great confusion in the sample size, for example for albacore tuna table S4 reports 44 samples but the sampling was carried out by taking 3 units (assumed to be of the same product) where possible (see line 97). This count leads to altered values that do not correspond to a variety of different samples. In my opinion the sample size for some species (oysters, squid, clams) is really too low to be able to use average values.
This has been clarified in the methods: “Products were purchased in groups of 3 or more (n = 3-44)…”
Regarding other results on similar products, among the references cited in the manuscript, only the one by Karimi et al., 2012 is considered in the tableS9.
This article was specifically used as a reference because it is the most comprehensive curation of mercury concentration data in commercial fish and shellfish, therefore it takes into account data across multiple studies.
Figure 2 brings together the comparison between brands (A and B) and different types of products and this generates confusion.
We apologize if this was initially confusing. Hopefully the new layout is more transparent and creates less confusion.
Figure 3 represents the trend of mercury concentrations detected by the FDA to which the result of the study is improperly added.
We are unsure about what exactly the reviewer means by improperly added. We made sure to only use FDA data that reported on total Hg, similar to our analysis, and although the two studies used different analytical techniques other groups (including in the EHP Karimi et al paper) have combined Hg data across studies by comparing grand means. In our approach we utilized a one-way ANOVA, which utilizes the grand mean to calculate between group sum of squares when comparing groups statistically.
Reviewer 3 Report
Comments and Suggestions for Authors
A technically sound study that however offers little novelty, and replicates long known knowledge. Sounds as a technical report to a local authority rather than a scientific paper. What do we learn from this study that we didn't already know? However, if the journal finds it suitable for their editorial policy, fine for me.
This said, I have only few technical comments.
Give the scientific names of the analyzed species
Lines 135-136: I certainly do not want to challenge your English, but I am puzzled by the sentence starting with "Where non-detects,..." - words missing?
Lines 158-161: it is not clear if (and how) you accounted for the different analytical protocols when comparing your data with previous FDA analyses. It is also not clear if you included analytical errors in the evaluation of the statistical significance.
Figures have a sort of heading statement (e.g, Fig. 3, tHG concentrations in this study are comparable to those previously reported by the FDA). Personally I have no objection, but this is quite unusual.
I do not want to question your statistics, but I am puzzled that 2022 values for albacore (range 0.2-1.27, mean 0.4) are considered significantly higher than 2004 (range 0.03-0.7, mean 0.31) but not 2005 (range 0.002-0.51, mean 0.33). The dataset for 2007 consists of only three analyses. Considering the previous comment at lines 158-161, I'd just say that overall the 2022 values are consistent with previous FDA studies.
Author Response
A technically sound study that however offers little novelty, and replicates long known knowledge. Sounds as a technical report to a local authority rather than a scientific paper. What do we learn from this study that we didn't already know? However, if the journal finds it suitable for their editorial policy, fine for me.
This said, I have only few technical comments.
Give the scientific names of the analyzed species
These have been added.
Lines 135-136: I certainly do not want to challenge your English, but I am puzzled by the sentence starting with "Where non-detects,..." - words missing?
This has been modified to say: “Where non-detects were reported, in other words Hg concentrations below the detection level of 0.01 PPM, we assigned an interpolated concentration of LOQ/√2 (Supplemental Tables S2 and S3).”
Lines 158-161: it is not clear if (and how) you accounted for the different analytical protocols when comparing your data with previous FDA analyses. It is also not clear if you included analytical errors in the evaluation of the statistical significance.
Figures have a sort of heading statement (e.g, Fig. 3, tHG concentrations in this study are comparable to those previously reported by the FDA). Personally I have no objection, but this is quite unusual.
The authors have done this in previous publications as a way to communicate the overall interpretation of the data presented in each figure.
I do not want to question your statistics, but I am puzzled that 2022 values for albacore (range 0.2-1.27, mean 0.4) are considered significantly higher than 2004 (range 0.03-0.7, mean 0.31) but not 2005 (range 0.002-0.51, mean 0.33). The dataset for 2007 consists of only three analyses. Considering the previous comment at lines 158-161, I'd just say that overall the 2022 values are consistent with previous FDA studies.
This has been modified as a result of modifying our analyses to log10 transformed data, where now we do see significant differences between 2022 and 2004, 2005, and 2007.
Reviewer 4 Report
Comments and Suggestions for Authors
In this study, the authors investigated the variability of mercury concentrations across species, brand, and tissue type in processed commercial seafood products. They reported that Hg concentrations are variable within and across species and brands, emphasizing the need for continued monitoring of the industry. The reviewer thinks that the results presented here are beneficial to mitigate consumer risks associated with consumption of process commercial fish products. The reviewer’s comments are given below.
Abstract: The abstract should be a total of about 200 words maximum (see “Manuscript Preparation of Toxics”).
Line 24: The significant figures of Hg concentrations shown in the text and Tables S1 and S4 are too many. They are at most three based on analytical error.
Lines 107 and 116: Correct “2, 2 ml” and “01.”.
Lines 133–135: In this study, the authors verified the measurement of Hg concentration using a NIST standard reference material (lines 119–122). Do the authors point out problems in Hg measurement in the FDA study?
Lines 149, 151 and 155: Explain abbreviations of “tHg DW”, “DW tHg” and “WW tHg”.
Lines 294–296: Is this an opinion based on any experimental evidence? If so, cite relevant literature.
Author Response
In this study, the authors investigated the variability of mercury concentrations across species, brand, and tissue type in processed commercial seafood products. They reported that Hg concentrations are variable within and across species and brands, emphasizing the need for continued monitoring of the industry. The reviewer thinks that the results presented here are beneficial to mitigate consumer risks associated with consumption of process commercial fish products. The reviewer’s comments are given below.
Abstract: The abstract should be a total of about 200 words maximum (see “Manuscript Preparation of Toxics”).
The abstract has been trimmed significantly, now 200 words.
Line 24: The significant figures of Hg concentrations shown in the text and Tables S1 and S4 are too many. They are at most three based on analytical error.
This has been fixed throughout.
Lines 107 and 116: Correct “2, 2 ml” and “01.”.
This has been corrected.
Lines 133–135: In this study, the authors verified the measurement of Hg concentration using a NIST standard reference material (lines 119–122). Do the authors point out problems in Hg measurement in the FDA study?
Although a NIST standard was not reported by the FDA for this analysis, they did utilize the same approach to generate their standard curve for ICP-MS using HgCl2 liquid standard.
Lines 149, 151 and 155: Explain abbreviations of “tHg DW”, “DW tHg” and “WW tHg”.
This has been clarified. See lines 157 – 160 and 207 – 209.
Lines 294–296: Is this an opinion based on any experimental evidence? If so, cite relevant literature.
Citations have been added.